# Copper-catalyzed carbo-difluoromethylation of alkenes via radical relay

Aijie Cai[1], Wenhao Yan[1], Xiaojun Zeng[1], Samson B. Zacate[2], Tzu-Hsuan Chao[3], Jeanette A. Krause[1], Mu-Jeng Cheng [3] & Wei Liu [1✉]

Organic molecules that contain alkyl-difluoromethyl moieties have received increased attention in medicinal chemistry, but their synthesis in a modular and late-stage fashion remains challenging. We report herein an efficient copper-catalyzed radical relay approach for the carbo-difluoromethylation of alkenes. This approach simultaneously introduces $CF_2H$ groups along with complex alkyl or aryl groups into alkenes with regioselectivity opposite to traditional $CF_2H$ radical addition. We demonstrate a broad substrate scope and a wide functional group compatibility. This scalable protocol is applied to the late-stage functionalization of complex molecules and the synthesis of $CF_2H$ analogues of bioactive molecules. Mechanistic studies and density functional theory calculations suggest a unique ligand effect on the reactivity of the $Cu-CF_2H$ species.

[1] Department of Chemistry, University of Cincinnati, Cincinnati, OH 45221, USA. [2] Department of Chemistry and Biochemistry, Miami University, Oxford, OH 45056, USA. [3] Department of Chemistry, National Cheng Kung University, Tainan 701, Taiwan. ✉email: liu2w2@uc.edu

The rapid construction of fluorine-containing organic molecules has attracted considerable attention, owing to the special ability of a fluorine atom or a fluoroalkyl group to modulate the metabolic stability, lipophilicity, and membrane permeability of a drug candidate[1-5]. Among the many fluorine-containing groups, the difluoromethyl group ($CF_2H$) is drawing ever-increasing attention due its unique ability to act as a possible lipophilic hydrogen bond donor and thus a potential bioisostere for hydroxyl, amino, or thiol groups[6-20]. It is not surprising, therefore, that pharmaceutical companies began to investigate $CF_2H$-containing lead compounds and intense research has been focused on the development of efficient methods for the installation of $CF_2H$ groups. Over the past decade, significant progress has been made in the subject of difluoromethylation of arenes[21-37]. However, analogous methods for the synthesis of alkyl-difluoromethanes have lagged behind, despite the high pharmaceutical relevance of these molecules (Fig. 1a).

Vicinal carbo-difluoromethylation of widely available alkenes, which results in the simultaneous formation of an alkyl–$CF_2H$ bond and a C–C bond, is an attractive, efficient, and potentially modular method for the synthesis of complex alkyl-difluoromethanes. Elegant work has recently been conducted by Dolbier[38], Qing[39], Xiao[40], Gouverneur[41], Zhang[42], Chu[43], and others[6,44] for the difluoromethylation of alkenes by harnessing the reactivity of an in situ generated difluoromethyl or difluoroalkyl radical. Although these state-of-the-art methods allow for the rapid difluoromethyl-functionalization of alkenes, nonetheless, in these reactions the $CF_2H$ group is generally attached to the terminal side of the alkenes, as dictated by the $CF_2H$ radical addition to the less sterically hindered site. In addition, the groups installed to the remaining site of the alkenes are limited to either hydrogen atoms or other simple functional groups (Fig. 1b). Therefore, an alternative strategy allowing for the introduction of a diverse range of groups onto alkenes with complementary regioselectivity would open an avenue for the discovery of $CF_2H$-pharmaceuticals.

Our group has recently demonstrated that copper can catalyze the efficient transfer of a $CF_2H$ group to an alkyl radical. This mode of $CF_2H$ transfer allowed us to develop the first decarboxylative[45], deaminative[46] and benzylic C–H difluoromethylation reactions[47]. These reactions proceed via copper-catalyzed generation of an alkyl radical which can be trapped by a [$Cu^{II}$-$CF_2H$] intermediate and go through reductive elimination to generate the desired alkyl-difluoromethane. We recently wondered whether this Cu/$CF_2H$ system could be applied to the difunctionalization of alkenes by intercepting the radical intermediate with a suitable alkene to form a relayed radical[48-54]. Subsequent copper-$CF_2H$ trapping/reductive elimination of the relayed radical could then allow for the carbo-difluoromethylation of the alkene. We anticipated that, by varying the radical precursors and alkene partners, this approach could enable the synthesis of complex alkyl-difluoromethanes in a modular fashion (Fig. 1c).

A plausible mechanism for the proposed carbo-difluoromethylation reaction is shown in Fig. 2. A transmetalation of a copper(I) catalyst 1 with a [$Zn^{II}$-$CF_2H$] reagent is expected afford a reactive [$Cu^{I}$-$CF_2H$] species 3[23,26,28,33,34,55], which should readily reduce a suitable alkyl or aryl electrophile 4, to form a carbon-centered radical 5. This radical species would then undergo a radical addition to an alkene 6, to generate the resultant relayed radical 7. The subsequent radical recombination of 7 with the [$Cu^{II}$-$CF_2H$] complex 8 could generate a formal $Cu^{III}$ intermediate 9[56], which undergoes reductive elimination to forge the desired alkyl-difluoromethane product 10. This reductive elimination step regenerates the $Cu^{I}$ catalyst and closes the catalytic cycle[57,58]. We recognized that a critical factor to the success of this reaction would be the selective addition of the

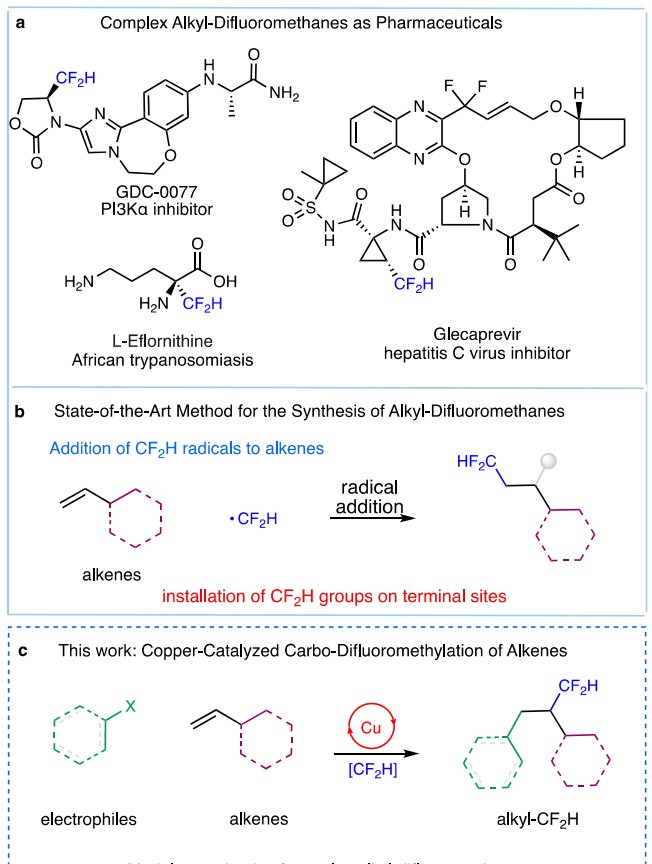

**Fig. 1 Copper-catalyzed carbo-difluoromethylation of alkenes could enable modular synthesis of complex alkyl-difluoromethanes. a** Complex alkyl-difluoromethanes as pharmaceuticals. **b** A state-of-the-art method for the synthesis of alkyl-difluoromethanes. **c** Copper-catalyzed carbo-difluoromethylaton of alkenes.

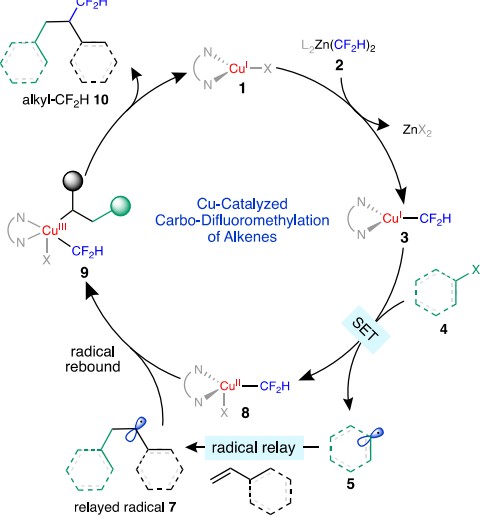

**Fig. 2 Proposed catalytic cycle for the copper-catalyzed carbo-difluoromethylation of alkenes.** The proposed mechanism involves the single electron transfer (SET) from a [Cu-$CF_2H$] species to an electrophile to form a carbon-centered radical, which reacts with an alkene to afford a relayed radical. The relayed radical reacts with a [$Cu^{II}$-$CF_2H$] species to form the difluoromethylated product.

**Fig. 3 Optimization of the copper-catalyzed difluoromethyl-alkylation of alkenes.** Reactions were conducted with **11** (0.1 mmol, 1.0 equiv.), styrene (0.3 mmol, 3.0 equiv.), (terpy)Zn(CF$_2$H)$_2$ (0.08 mmol, 0.8 equiv.) and CuCl (30 mol%) in DMSO (0.4 mL). (terpy)Zn(CF$_2$H)$_2$ was formed in situ by pre-mixing (DMPU)$_2$Zn(CF$_2$H)$_2$ and terpy in DMSO. Yields were determined by $^{19}$F NMR using 1-fluoro-3-nitrobenzene as the internal standard. terpy, 2,2′:6′,2″-terpyridine; DMSO, dimethyl sulfoxide; NHPI, N-hydroxyphthalimide; DMPU, N,N′-Dimethylpropyleneurea.

| Entry | Variation from standard conditions | Yield **12** |
|---|---|---|
| 1 | None | 76% (11%, 12a) |
| 2 | 2,2'-bipyridine instead of terpy | trace |
| 3 | 4,4'4''-tri-tert-butyl-terpy instead of terpy | 32% |
| 4 | without terpy | trace |
| 5 | use NHPI ester | trace |
| 6 | [Cu(CH$_3$CN)$_4$]•PF$_6$ instead of CuCl | 65% |
| 7 | Cu(acac)$_2$ instead of CuCl | 61% |
| 8 | 0.6 equiv. (terpy)Zn(CF$_2$H)$_2$ | 60% |
| 9 | 1.0 equiv. (terpy)Zn(CF$_2$H)$_2$ | 62% |
| 10 | reaction was run at 60 °C | 58% |
| 11 | 2.0 equiv. of styrene | 59% |
| 12 | no CuCl | N.D. |
| 13 | Outside of glovebox | 74% |

carbon-centered radical **5** to the alkenes in lieu of direct recombination to the [Cu$^{II}$-CF$_2$H] species. We expected that controlling the rates of radical recombination to the copper catalyst would be necessary to achieve the desired transformation. It is noteworthy that although difunctionalization of alkenes via a copper-catalyzed radical relay process—the concept of which was first conveyed by Liu and Stahl[59]—has been reported[60–63], the simultaneous formation of two Csp$^3$–Csp$^3$ bonds via this process remains rare[64].

In this work, we report a copper-catalyzed carbo-difluoromethylation reaction that can simultaneously introduce CF$_2$H groups and complex alkyl or aryl groups into alkenes with the regioselectivity opposite to traditional CF$_2$H radical chemistry.

## Results

**Difluoromethyl-alkylation of alkenes**. Based on our previous work on the decarboxylative difluoromethylation of redox-active esters (RAEs), we first questioned whether an alkyl radical generated in situ from RAEs could participate in this reaction, allowing for the development of the difluoromethyl-alkylation of alkenes. Thus, we began our studies with the RAE of tetrahydropyran-4-carboxylic acid **11** as the alkyl electrophile, with styrene as the alkene partner. Different copper salts, ligands, and CF$_2$H sources were evaluated (Fig. 3). To our delight, we found that the use of CuCl as the catalyst, along with (terpy)Zn(CF$_2$H)$_2$ (terpy = 2,2′:6′,2″-terpyridine) as the CF$_2$H source, generated via mixing terpy and (DMPU)$_2$Zn(CF$_2$H)$_2$ in situ, the product **12** could be formed in a 76% yield in dimethyl sulfoxide (DMSO) at room temperature (r.t.) (entry 1). The formation of **12** represents a rare example of the simultaneous formation of two Csp$^3$–Csp$^3$ bonds via a copper-catalyzed radical relay process. In addition, the side product **12a**, derived from the direct

trapping of the alkyl radical by the [Cu$^{II}$-CF$_2$H] species, was formed in a 11% yield under the optimized conditions. Intriguingly, the ligand used in the reactions plays a vital role in this transformation. No desired products were formed when styrene was added to our previously established decarboxylative difluoromethylation conditions, in which 2,2′-bipyridine (bpy) was used as the ligand (entry 2). Other bidentate ligands provided essentially no products (Supplementary Table 1) and lower yields were observed when other substituted terpy ligands were used instead of terpy (entry 3). No desired product was formed when (DMPU)$_2$Zn(CF$_2$H)$_2$ was used as the difluoromethyl source without the addition of terpy (entry 4). Trace amounts of products were formed when the analogous N-hydroxyphthalimide (NHPI) ester was used (entry 5). Different copper(I) and copper (II) salts could also be used (entries 6 and 7 and Supplementary Table 2), albeit with lower efficiency. Varying the amounts of (terpy)Zn(CF$_2$H)$_2$ and increasing the reaction temperature led to diminished yields (entries 8–10). The styrene loading can be reduced to 2 equiv., yet attaining a synthetically useful yield (entry 11). No difluoromethylated products were formed when the reactions were conducted in the absence of copper catalysts (entry 12). This reaction could also be set up outside of the glovebox without affecting the yield (entry 13 and Supplementary Method 1).

With the optimized conditions in hand, we began to evaluate the scope of vinyl arenes (Fig. 4). A wide range of electron-withdrawing, electron-donating, and electron-neutral functional groups could be tolerated on the aryl rings (**12–28**). The reactivity was not hindered by ortho substitutions on the vinyl arene ring (**16**). A basic tertiary amine group, which is usually problematic in transition metal catalysis, was tolerated (**18**). The compatibility

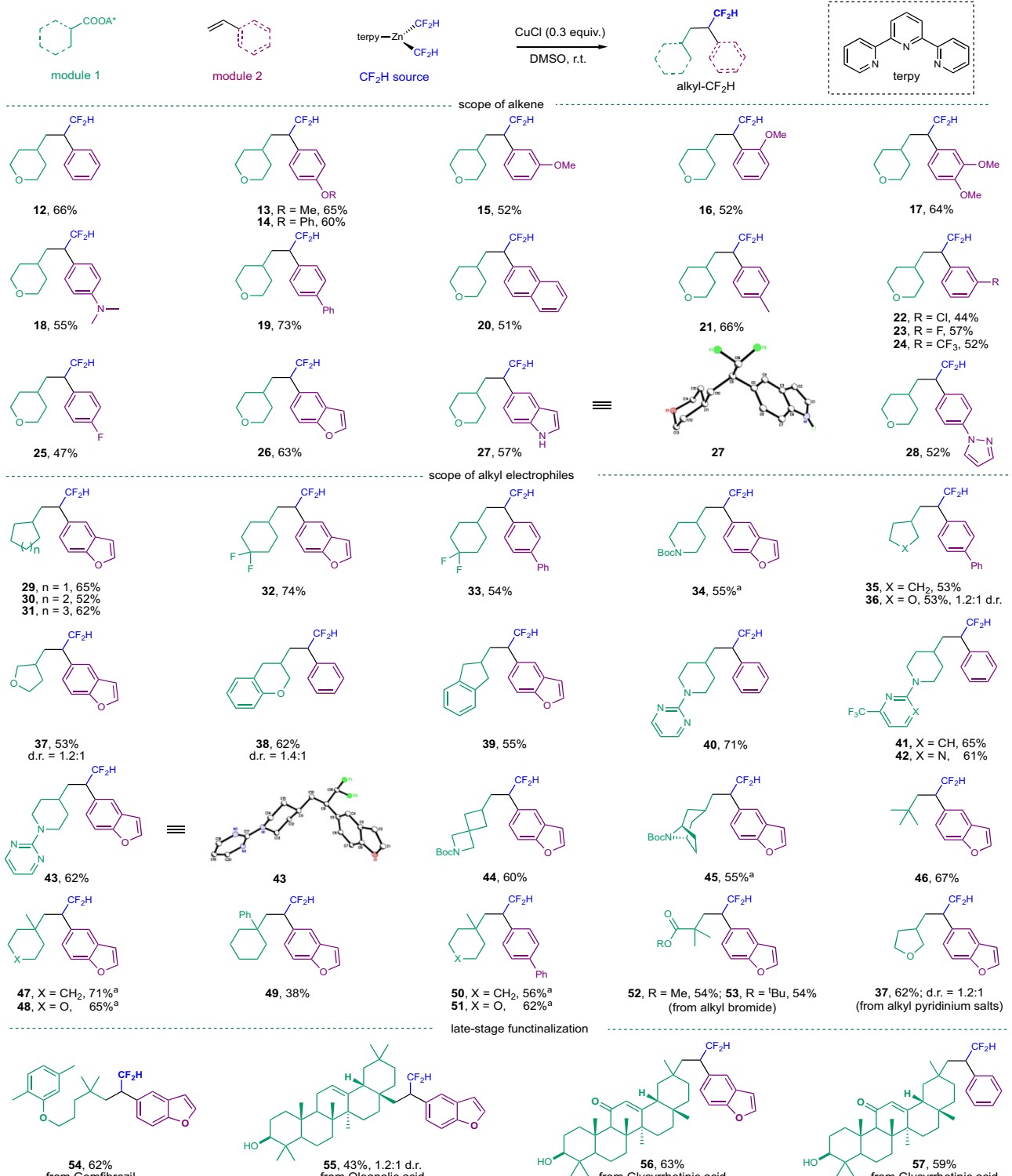

**Fig. 4 Copper-catalyzed difluoromethyl-alkylation of alkenes.** Reactions were run with 0.25 mmol of RAEs, 0.75 mmol of alkenes (3.0 equiv.), 0.2 mmol of (terpy)Zn(CF$_2$H)$_2$ (0.8 equiv.), and CuCl (30 mol%) in 1 mL of DMSO at r.t., isolated yields based on RAEs. [a]Reaction performed under modified conditions (Supplementary Method 5). terpy, 2,2′;6′,2″-terpyridine; DMSO, dimethyl sulfoxide.

with an aryl chloride (**22**) provides the basis for the further synthetic manipulation of the difluoromethylated products. This protocol allowed for the efficient installation of nitrogen- and oxygen-containing heterocycles, including benzofuran (**26**), unprotected indole ring (**27**), and pyrazole (**28**), which are ubiquitous in pharmaceuticals, into alkyl-CF$_2$H moieties in good yields. It is worth noting that the excess vinyl arenes could be

recovered with excellent mass balance (Supplementary Method 1).

We next turned our attention to the scope of carboxylic acids for this protocol. To our delight, this reaction was amenable to a wide range of secondary and tertiary carboxylic acids. RAEs derived from secondary carboxylic acids appended to a variety of four-, five-, six-, or seven-membered rings, paired with different

functional groups could be successfully employed to deliver the desired products in good to excellent yields (**29–45**). Oxygen-containing saturated heterocycles, such as tetrahydrofuran (**36, 37**) and tetrahydropyran (**38**), were successfully employed in this reaction. Carboxylic acids derived from piperidine, one of the most prevalent ring systems found in drug molecules[65], could undergo efficient difluoromethylation (**40–43**). Nitrogen-containing heterocycles, which are core structures in drug synthesis, including pyrimidine (**40, 42**) and pyridine (**41**), were all accommodated in this protocol. Two bicyclic and spirocyclic carboxylic acids were successfully converted to the desired products with good efficiency (**44, 45**). Furthermore, acyclic and cyclic tertiary carboxylic acids could undergo the radical relay pathway to give the desired products in good yields (**46–51**). Moderate yields were observed when a phenyl group was connected to the tertiary carboxylic acid (**49**), likely due to the lower reactivity of a tertiary benzylic radical. In addition to RAEs, activated alkyl bromides (**52, 53**) and an alkyl pyridinium salt (**37**) were found to participate readily in this reaction, providing the corresponding alkyl-CF$_2$H products in good yields. Poor yields were observed when the RAEs of primary carboxylic acids were employed in the reactions and the decarboxylative difluoromethylated products were found to be the dominant products.

To further showcase the synthetic utility of this carbo-difluoromethylation protocol for the construction of complex alkyl-difluoromethanes, we sought to apply it to the late-stage modification of pharmaceuticals and natural products. Modification of a lipid regulating agent, gemfibrozil, afforded the desired alkyl-difluoromethane **54** in a 62% yield. Furthermore, we undertook the rapid functionalization of two complex steroidal acids, glycyrrhetinic acid and oleanolic acid, both of which could be converted to the corresponding difluoromethylated products in synthetically useful yields (**55–57**). The fact that unprotected hydroxyl groups could be well tolerated in these transformations further highlighted the mild conditions of this copper-catalyzed process. Given the ubiquity of carboxylic acids and alkenes, we expect that this protocol will find wide application in the field of medicinal chemistry for the synthesis of complex CF$_2$H-containing pharmaceuticals in a modular fashion.

**Difluoromethyl-arylation of alkenes.** Having demonstrated that in situ generated alkyl radicals can participate in this carbo-difluoromethylation protocol, we further questioned whether aryl radicals could also be engaged in this reaction, allowing for the simultaneous introduction of an aryl and a CF$_2$H group to various alkenes. A possible source for aryl radicals could be aryl diazonium salts, which are readily prepared from various anilines. We reasoned that an aryl diazonium salt could undergo a homolytic bond cleavage in the presence of a reactive [Cu$^I$-CF$_2$H] species, forming an aryl radical[34], which could react with an alkene via the Meerwein-type arylation pathway. The relayed radical could be trapped by a [Cu$^{II}$-CF$_2$H] species, forming the desired difluoromethylated product via the radical recombination/reductive elimination pathway. It is noteworthy that although the Meerwein arylation reaction, which was originally reported by Hans Meerwein in 1939[66], has been developed as a useful method for the functionalization of alkenes[67], the interception of the relayed radical by a copper catalyst for the construction of a C–C bond remains elusive.

We have found that in the presence of [Cu(CH$_3$CN)$_4$]BF$_4$ (20 mol%) as the catalyst, terpy (20 mol%) as the ligand and (DMPU)$_2$Zn(CF$_2$H)$_2$ as the difluoromethyl source, various aryl diazonium salts could undergo efficient difluoromethylation with different alkenes as the radical acceptors (Fig. 5 and see

Supplementary Table 3 for optimization details). Aryl diazonium salts bearing electron-withdrawing or electron-donating groups reacted efficiently with vinyl arenes, affording the corresponding difluoromethylated products in good yields. Functional groups including halides (**61–63**), esters (**64**), ketones (**68**), and acetals (**68**) were well tolerated in these transformations. Ortho substitutions of the aryl diazonium salts did not hinder the reactivity (**63**). Aryl diazonium salts bearing heterocycles, such as dioxane (**67**), coumarin (**69**), and thioazole (**70, 71**), along with a heteroaryl diazonium salt (**72**) afforded the desired difluoromethylated products albeit in moderate yields. In addition to styrenes, other alkenes including acrylonitrile and methyl acrylate, could also be engaged in this difluoromethyl-arylation reaction, allowing for the rapid synthesis α-CF$_2$H nitriles and α-CF$_2$H esters (**73–82**). In addition, the difluoromethyl-arylation of acrylonitrile could also be conducted on a large scale. Without any modification of the standard conditions, the gram-scale synthesis of **76** could be produced in a similar yield to that obtained on a 0.25 mmol scale. Given the diverse reactivities of cyanide groups and ester groups, we expected that α-CF$_2$H nitriles and α-CF$_2$H esters could serve as highly useful building blocks for the preparation of other CF$_2$H-containing moieties. For example, the treatment of an α-CF$_2$H nitrile **76** with an alkaline solution of H$_2$O$_2$ afforded an α-CF$_2$H amide **83** in an 80% yield, while the reduction of **81** with diisobutylaluminium hydride (DIBAL-H) allowed for the synthesis of a β-CF$_2$H alcohol **84** in an 85% yield.

The synthetic utility of this difluoromethyl-arylation reaction has been further highlighted by the rapid synthesis of the CF$_2$H analogue of combretastatin, **85**, in a 45% yield. Combretastatin and its derivatives are potent vascular targeting agents and are undergoing various clinical trials for treating different cancers[68]. Considering the known ability of a CF$_2$H group to act as a bioisostere of a hydroxyl group, **85** could potentially be a metabolically stable analogue of combretastatin.

**Mechanistic studies.** Mechanistic studies were conducted to shed light on these carbo-difluoromethylation reactions (Fig. 6). In our previously reported decarboxylative difluoromethylation reactions, bpy was found to be the optimal ligand[45], but the tri-dendate ligand, terpy, was uniquely effective for the current system. We were intrigued by the dramatic difference of the difluoromethylation reactivity with difference ligands. Mechanistic understanding of the effect of ligands on the difluoromethylation reactivity is vital for the development of other alkyl-CF$_2$H bond-forming reactions and could shed light on copper-catalyzed C–C bond-forming reactions that involve alkyl radicals as intermediates.

When the carbo-difluoromethylation reaction of **86** was conducted with bpy as the ligand, trace amount of the desired product **87** was observed. Analysis of the reaction mixture by GC/MS showed that the dimerization of the relayed radical (**88**) was a major competing pathway (Fig. 6a and Supplementary Figs. 11–12). We hypothesized that the reason for this dramatic difference in reactivity with different ligand is that the relayed radical, a secondary benzylic radical, could smoothly react with the [(terpy)Cu$^{II}$CF$_2$H] intermediate but could not be efficiently trapped by a [(bpy)Cu$^{II}$CF$_2$H] species.

Several additional experimental results further support this hypothesis. When a RAE derived from a secondary benzylic carboxylic acid, **89**, was subjected to the standard decarboxylative difluoromethylation conditions, trace amount of the difluoromethylated product **90** was detected with bpy as the ligand, a phenomenon we observed in our previous studies[45]. On the contrary, a nearly quantitative yield was observed for the

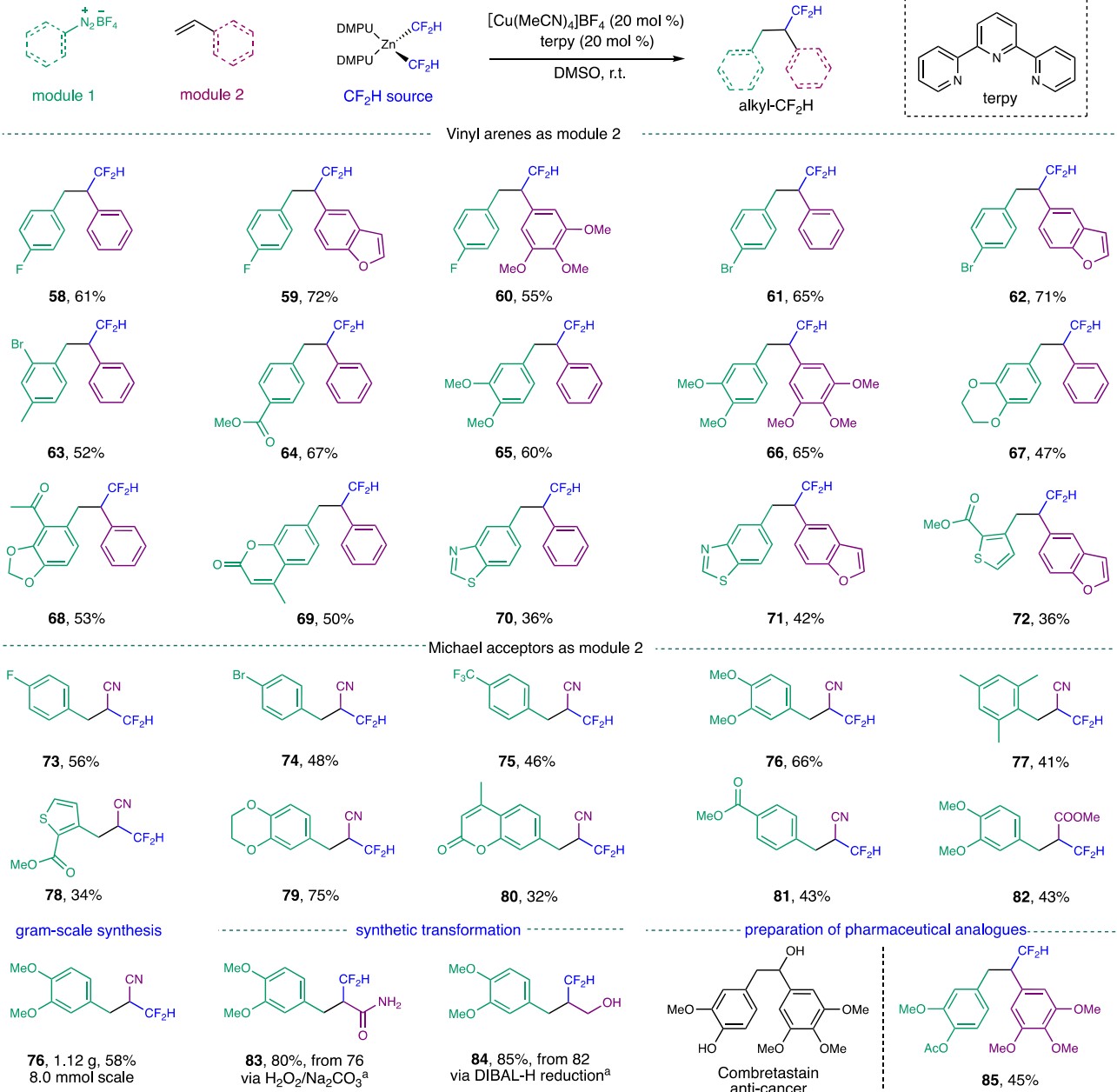

**Fig. 5 Copper-catalyzed difluoromethyl-arylation of alkenes.** Reactions were run with 0.25 mmol of aryl diazonium salts (1.0 equiv.), 0.75 mmol of alkenes (3.0 equiv.), 0.2 mmol of (DMPU)$_2$Zn(CF$_2$H)$_2$ (0.8 equiv.), CuCl (20 mol%), and terpy (20 mol%) in 1 mL of DMSO at r.t., isolated yields. [a]See Supplementary Method 1 for detailed conditions. terpy, 2,2′;6′,2″-terpyridine; DMSO, dimethyl sulfoxide.

formation of **90** when terpy was used as the ligand. Similarly, the RAE derived from an acyclic secondary carboxylic acid **91** could undergo efficient decarboxylative difluoromethylation with terpy as the ligand, forming **92** as the major product, while a poor reactivity was observed with a bpy ligand under otherwise identical conditions (Fig. 6b). More importantly, in both cases, the dimerization of the secondary alkyl radicals was the competing reaction when bpy was used as the ligand (Supplementary Figs. 5–10).

To take into account of an alternative possibility that this ligand effect is due to the differences in reactivity of the [Cu$^I$-CF$_2$H] species to undergo a single electron transfer event, we conducted the decarboxylative difluoromethylation of a radical clock substrate **93** (Fig. 6c). Direct decarboxylative difluoromethylation of **93** with either terpy or bpy as ligands under otherwise identical conditions both afforded the ring-opened

product **94** in high yields along with trace amounts of the unrearranged product **95**. Although further experiments are necessary for the detailed comparison of the reactivity of different [Cu$^I$-CF$_2$H] species, these results support that both [(terpy)Cu$^I$-CF$_2$H] and [(bpy)Cu$^I$-CF$_2$H] species were reactive enough to undergo the single electron transfer events with the RAE **93** to generate the corresponding benzylic radical, which rearranged to a primary alkyl radical, affording **94** as the major product. The high yields of **94** with either ligand could be explained by the small steric interaction when a primary radical reacts with the [Cu$^{II}$-CF$_2$H] species.

**DFT calculation on the effect of ligands**. Our mechanistic studies suggest that acyclic secondary alkyl radicals could be more efficiently trapped by the [Cu$^{II}$-CF$_2$H] species coordinated to a

**a.** Dimerization of the relayed radical with bpy as the ligand

**b.** Effect of ligands on the decarboxylative difluoromethylation reactions

**c.** Effect of ligands on the decarboxylative difluoromethylation of a radical clock

**Fig. 6 Mechanistic studies. a** Dimerization of the relayed radical with bpy as the ligand. **b** Effect of ligands on the decarboxylative difluoromethylation reactions. **c** Radical clock experiments. terpy, 2,2′;6′,2″-terpyridine; bpy, 2,2′-bipyridine; DMSO, dimethyl sulfoxide.

terpy ligand. To better understand this ligand effect, we performed DFT calculations to probe the reaction between a secondary benzylic radical and the [$Cu^{II}$-$CF_2H$] species bound with a bpy or a terpy ligand (Fig. 7). Our previous DFT calculations[69] and Cook's work[70] on the trifluoromethylation of benzylic C–H bonds have both indicated that the benzylic radical could recombine with a four-coordinate [(bpy)$Cu^{II}$($CF_3$)$_2$] species to form a five-coordinate high-valent copper(III) intermediate, [(bpy)$Cu^{III}$($CF_3$)$_2$(benzyl)], which reductively eliminates to form the trifluoromethylated product. Very recently, the Shen group has successfully characterized the putative five-coordinate copper (III) complex, [(bpy)$Cu^{III}$($CF_3$)$_2$(Me)], which could undergo C–$CF_3$ bond-forming reductive elimination to form 1,1,1-trifluoroethane[58]. These precedents suggest that $Csp^3$-fluoroalkyl bond-forming reductive elimination would most likely occur on five-coordinate copper(III) complexes.

With this background in mind, we reasoned that when bpy was used as the ligand for the difluoromethylation reaction, a benzylic radical would react with a four-coordinate copper(II) intermediate which bound to two $CF_2H$ groups, [(bpy)$Cu^{II}$($CF_2H$)$_2$] (**Cu-2**, Fig. 7), to form a five-coordinate copper(III) complex (**Cu-2A**), which reductively eliminated to form the difluoromethylated product. On the contrary, when the tridentate ligand, terpy, bound with the copper center, the four-coordinate copper(II)

could only attach to one $CF_2H$ group (**Cu-1**). The recombination of the benzylic radical with **Cu-1** could form the five-coordinate high-valent copper(III) complex (**Cu-1A**), with only one $CF_2H$ bonding to the copper center.

Interestingly, our DFT calculation found that the formation of the high-valent $Cu^{III}$ species (**Cu-1A**) from the combination of a benzylic radical (**Rad**) with **Cu-1** was a barrierless pathway. In contrast, the reaction between **Rad** with **Cu-2** required a high activation energy ($\Delta G^{\ddagger} = 21.8$ kcal/mol, **TS2**), suggesting that the combination of the secondary benzylic radical, **Rad**, with **Cu-1** was a much more efficient process than its reaction with **Cu-2**. It is noteworthy that **Cu-2** was found to combine with a cyclohexyl radical via a barrierless pathway (Supplementary Fig. 13), consistent with our previous experimental results that the Cu/bpy system was effective for the decarboxylative difluoromethylation of the RAEs derived from cyclic secondary carboxylic acids.

Moreover, our calculation results demonstrated that the C–$CF_2H$ bond-forming reductive elimination step also contributed to the effect of ligands. As shown in Fig. 7, the reductive elimination of **Cu-1A** proceeded with a significantly lower activation energy ($\Delta G^{\ddagger} = 6.3$ kcal/mol, **TSre-1**) than that of **Cu-2A** ($\Delta G^{\ddagger} = 18.1$ kcal/mol, **TSre-2**). Overall, these DFT calculations suggest that the success of the tridentate ligand, terpy, in the

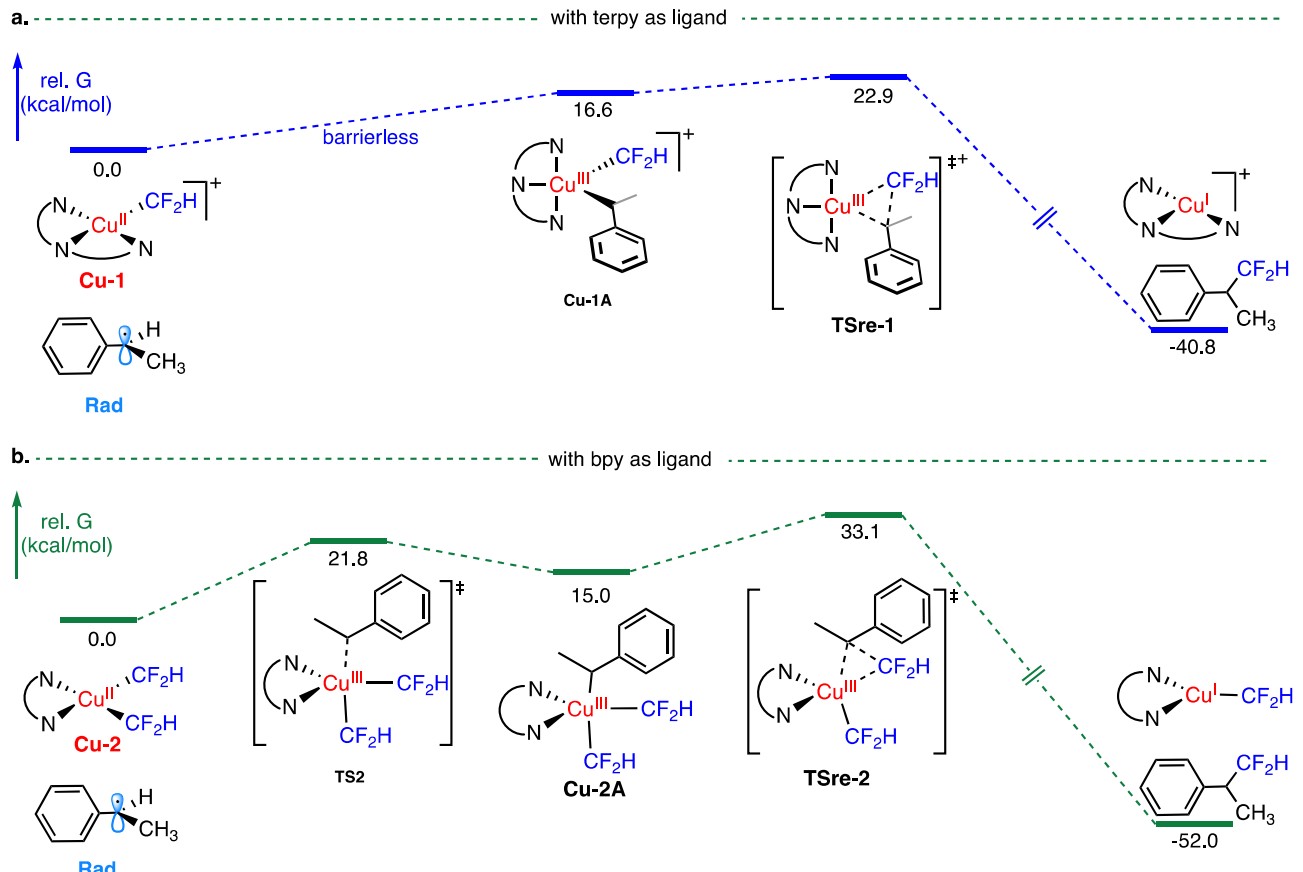

**Fig. 7 Energy profile of the reactions between a secondary benzylic radical with [Cu$^{II}$-CF$_2$H] species. a** Energy profile of the reactions between a secondary benzylic radical with [(terpy)Cu$^{II}$(CF$_2$H)] species, **b** Energy profile of the reactions between a secondary benzylic radical with [(bpy)Cu$^{II}$(CF$_2$H)$_2$] species. Rad, benzylic radical; terpy, 2,2′;6′,2″-terpyridine; bpy, 2,2′-bipyridine; TS, transition state; TSre, transition state for reductive elimination; rel. G, relative Gibbs free energy.

carbo-difluoromethylation reactions was largely due to a rapid combination of the secondary benzylic radicals with the Cu$^{II}$ intermediates as well as the facile reductive elimination of the Cu$^{III}$ species, although other factors including the difference of the transmetalation rates could not be ruled out.

## Discussion

In conclusion, we reported herein a carbo-difluoromethylation of alkenes via a copper-catalyzed radical relay strategy, complementing the well-established CF$_2$H radical addition to alkenes. This transformation allows for the rapid installation of CF$_2$H groups and a diverse range of alkyl and aryl groups onto various vinyl arenes and Michael acceptors. This protocol could enable the efficient construction of complex alkyl-difluoromethanes in a modular fashion and rapid evaluation of structure–activity relationships of potential CF$_2$H-containig pharmaceuticals. Furthermore, the difluoromethyl-alkylation protocol represents a rare example of a copper-catalyzed radical relay process for the simultaneous construction of two Csp$^3$–Csp$^3$ bonds, while the difluoromethyl-arylation reaction could open an avenue for the development of other Meerwein-type arylation reactions. Mechanistic studies indicated a previously unknown effect of ligands on the reactivity of the putative [Cu-CF$_2$H] species, which may serve as a guide in selecting ligands in other copper-catalyzed reactions which involve alkyl radicals as the intermediates. Further studies on the use of other electrophiles/radical acceptors and the asymmetric version of this reaction are currently ongoing in our laboratories.

## Methods

**Difluoromethyl-alkylation of alkenes**. In a nitrogen-filled glovebox, to a 4 mL vial equipped with a stir bar was added terpyridine (0.8 equiv., 47 mg), (DMPU)$_2$Zn (CF$_2$H)$_2$ (0.8 equiv., 87 mg), and 800 μL DMSO. The resulting mixture was stirred at room temperature for 1 min to generate (terpy)Zn(CF$_2$H)$_2$ in situ. A different 4 mL vial equipped with a stir bar was sequentially charged with CuCl (30 mol%, 7.5 mg), RAEs (0.25 mmol, 1.0 equiv.), the DMSO solution of the in situ formed (terpy)Zn(CF$_2$H)$_2$, and alkene (0.75 mmol, 3.0 equiv.) in DMSO (200 μL). The resultant mixture was stirred at room temperature for 12 h. After the reaction was completed, the mixture was diluted with ethyl acetate (50 mL), filtered through a short plug of Celite, and washed with H$_2$O (50 mL) and brine. The organic layer was combined, dried over Na$_2$SO$_4$, filtered, and then concentrated under reduced pressure. The crude product was purified by flash column chromatography.

**Difluoromethyl-arylations of alkenes**. In a nitrogen-filled glovebox, a 4 mL vial equipped with a stir bar was charged with [Cu(MeCN)$_4$]BF$_4$ (20 mol%, 15.5 mg), terpyridine (**L1**) (20 mol%, 11.8 mg), alkenes (0.75 mmol, 3.0 equiv.), and 200 μL DMSO. A solution of diazonium salts (0.25 mmol, 1.0 equiv.) in 400 μL DMSO, and a solution of (DMPU)$_2$Zn(CF$_2$H)$_2$ (0.2 mmol, 87 mg, 0.8 equiv.) in 400 μL DMSO were slowly added to the vial with syringes at the same time over the course of 10 min. The resultant mixture was stirred at room temperature for 30 min. After the reaction was completed, the mixture was diluted with ethyl acetate (50 mL), filtered through a short plug of Celite, and washed with H$_2$O (50 mL) and brine. The organic layer was combined, dried over Na$_2$SO$_4$, filtered, and then concentrated under reduced pressure. The crude product was purified by flash column chromatography.

## Data availability

The authors declare that all data supporting the findings of this study are available within the article and its supplementary information file, and also are available from the corresponding author upon reasonable request. The experimental procedures and characterizations of all new compounds are provided in Supplementary Information. The X-ray crystallographic coordinates for structures reported in this study have been

deposited at the Cambridge Crystallographic Data Centre (CCDC), under deposition numbers CDCC 2033193 (27), and CDCC 2033194 (43). These data can be obtained free of charge from www.ccdc.cam.ac.uk/data_request/cif.

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

## Acknowledgements

W.L. thanks University of Cincinnati for financial support. Funding for the D8 Venture diffractometer was through NSF-MRI grant CHE-1625737. NMR experiments were performed using a Bruker AVANCE NEO 400 MHz NMR spectrometer (funded by NSF-MRI grant CHE-1726092). We also thank Prof. John T. Groves and Mr. Yuan Cao (Princeton University) for the assistance with the high-resolution mass spectroscopy.

## Author contributions

W.L. and A.C. conceived the project and designed the experiments. A.C., W.Y., X.Z. and S.Z. developed the reactions, and contributed to the reaction scope. J.K. solved the crystal structures. M.C. and T.C. performed the theoretical calculation. W.L. and A.C. wrote the paper, supplementary information, and related materials.

## Competing interests

The authors declare no competing interests.
