## [Peer Review File · Nature Communications]

Reviewers' Comments:

Reviewer #1:

Remarks to the Author:

Liu and coworkers described a copper-catalyzed carbon-difluoromethylation of styrene derivatives via a radical relay strategy. The reaction conditions were systematically investigated in terms of the copper sources, the solvent, the temperature and more important, the ligands. Terpy was identified as the key ligand for the success of such a radical relay process since the formation of the desired difunctionalized products was not observed when a bidentate ligand bpy was used. Not only sp³-hybridized carbon radical but also sp²-hybridized carbon radical were successfully employed in the transformation. A gram-scale reaction was conducted. Initial mechanistic investigation on the effect of the terpy ligand was conducted and DFT calculation showed that the barrier for the radical recombination with terpy-ligated copper(II) species were much lower than that of bpy-ligated copper(II) species, which confirms the importance of the choice of the ligand.

The manuscript was well-written and related references were well-cited.

Nevertheless, there are some problems for the characterization of the compounds. For example, the HRMS data of compounds 12, 15, 22-24, 51-52, 54, 58, 60-70, 72, 80, 84-85 were missing. In addition, the measured HRMS data for compound 41 did not match with the calculated data.

Mechanistically, the first step of the catalytic reaction is the electron-transfer from [(Terpy)Cu(CF₂H)] to tetrahydropyran-4-carboxylic acid to generate Cu(II) species and a radical anion which undergoes decarboxylation to give alkyl radical. Is it possible that the electron-transfer step to [(Terpy)Cu(CF₂H)] is more efficient than that to [(bpy)Cu(CF₂H)₂]?

Considering the importance of the difluoromethyl group for the medicinal chemists and the field of pharmaceutical/organochemical industry, I suggest to accept it with minor revision.

Reviewer #2:

Remarks to the Author:

Dear Editor,

The authors describe a very interesting approach to a Cu-catalyzed carbo-difluoromethylation of various alkenes, which is in fact, a type of three-component reaction that involves two sequential Csp³-Csp³ bond formation initiated by a regioselective radical relay from secondary/tertiary alkyl radical to the alkene substrate. The driving force for the regioselectivity of this process is the formation of the most stable relayed radical, such as for instance the benzylic one, and the use of the appropriate ligands for the catalyst. The mechanistic conclusions are supported by both experimental and computational studies (DFT). Considering the complexity of the reaction (two C-C bond formations), its regioselectivity and the broad substrate scope and functional groups tolerance (including amine and hydroxyl groups), the yields of the large number of products synthesized are impressive. Therefore, this difluoromethylation may be useful for the synthesis of various carbo-CF₂H-containing derivatives, relevant for medicinal chemistry, in good yields. The manuscript is well organized and written. Based on the efficiency of this unique process and its high potential for the synthesis of new CF₂H-containing bioactive compounds, I suggest publication in Nature Communications subject to some minor revisions.

Comments:

1. In the introduction (page 1 line 23) regarding the statement that CF₂H "act as a lipophilic hydrogen bond donor and thus a bioisostere for hydroxyl, amino, or thiol groups", more accurately, it should be noted that for thiol, both groups, i.e. SH and CF₂H exhibit similar lipophilicities and hydrogen bond donating capabilities (J. Med Chem. 2019, 62, 5628-5637). There is a lack of appropriate references and examples for the increased implementation of this bioisosterism in medicinal chemistry. As in most cases, this is actually the rationale for the increased interest in the development of new synthetic approaches for difluoromethylation, and therefore, it is important to add a few of the recent reports on the utilization of the CF₂H group as a H-bond donor in drug discovery (see for instance ACS Med. Chem. Lett. 2018, 9(2), 143-8.; J.

Med. Chem. 2018, 61, 10084-105; Eur. J. Med. Chem. 2018, 143, 473-90.; Steroids 2019, 151, 108469.

2. Page 6 Table 2: a) the molecular structures as well as the font size of the atoms should be increased (the sizes in fig 4 are good enough). b) The diastereomeric ratio of compound 37 is missing. According to its spectrum in the SI it also seems to be of ca 1.2:1. c) More importantly, what is the rational for the synthesis of compounds 54-57 in terms of drug design (or discovery)? These transformations, which describe a replacement of the carboxylic acid group in several drugs, by the aryl-difluoromethyl group, does not fall in any category of structure activity relationship studies nor in bioisosterism study. Thus the rational for the synthesis of these specific modified drugs should be given.

3. Page 8 Table 3: the molecular structures as well as the font size of the atoms should be increased as mentioned above. The rational for the replacement of the hydroxyl group in Combretastain by the CF₂H group is clear in terms of bioisosterism and is a very nice example for the potential implementation of the new reaction.

4. Page 12, Methods: Can the reaction be successfully performed only in a glovebox? If simple Schlenck cannot be used, this seems as a drawback of the reaction, as most organic chemists do not use glovebox for their syntheses.

Thank You

Reviewer #3:

Remarks to the Author:

Liu and coworkers present an interesting three component difluoromethylation of alkenes for the efficient synthesis difluoroalkylation products, which was achieved by copper catalyzed radical relay process. Compared to the extensively study on the difluoromethyl radical addition to alkenes, the current reaction proceeds a novel carbon radical coupled with Cu(II)-CHF₂ to generate sp³-sp³ C-C bond. As described in manuscript, broad substrate scope was observed to give products with moderate to good yields. More interesting, the ligand effect on the radical coupling was investigated from experimental and DFT calculation, and ligand terpy exhibited much better reactivity than bpy, where the reaction of cationic (terpy)Cu(II)CHF₂ with radical have lower energy barrier than neutral (bpy)Cu(II)CHF₂. Overall, this is a very nice study in the field, and the current manuscript is deserved to be publication in Nature Communication with minor revision.

Comments:

(1) For the optimizing reaction condition, the full picture of the reaction should be provided. For instance, based on the reaction in Fig. 3B, the alkyl radical generated from decarboxylation is possibly trapped by (terpy)Cu(II)CHF₂. Thus, the side product should be addressed in Table 1, which is helpful to readers.

(2) For the ligand effect, another possibility should be discussed, where the transmetallation step of Cu(I) with LZn(CHF₂)₂ might also affect by ligand.

(3) For the reaction in Fig. 3C, the reaction gave the similar yields with terpy and bpy. wherein the primary carbon radical was involved. The possible reason should be given in the main text.

Point by point responses

Reviewer: 1

Liu and coworkers described a copper-catalyzed carbon-difluoromethylation of styrene derivatives via a radical relay strategy. The reaction conditions were systematically investigated in terms of the copper sources, the solvent, the temperature and more important, the ligands. Terpy was identified as the key ligand for the success of such a radical relay process since the formation of the desired difunctionalized products was not observed when a bidentate ligand bpy was used. Not only sp^3 -hybridized carbon radical but also sp^2 -hybridized carbon radical were successfully employed in the transformation. A gram-scale reaction was conducted. Initial mechanistic investigation on the effect of the terpy ligand was conducted and DFT calculation showed that the barrier for the radical recombination with terpy-ligated copper(II) species were much lower than that of bpy-ligated copper(II) species, which confirms the importance of the choice of the ligand.

The manuscript was well-written and related references were well-cited.

Our response: We thank the referee for the valuable time spent evaluating our work and the very supportive comments.

Nevertheless, there are some problems for the characterization of the compounds. For example, the HRMS data of compounds 12, 15, 22-24, 51-52, 54, 58, 60-70, 72, 80, 84-85 were missing. In addition, the measured HRMS data for compound 41 did not match with the calculated data.

Our response: We thank the referee for carefully evaluating the supporting information and pointing out these issues. High-resolution mass data has been provided for all of these compounds in the revised supporting information. Some of the CF_2H -containing compounds did not show the parent peaks in the high-resolution mass spectroscopy, and the peaks for the loss of HF were dominant (compounds **12**, **22-24**, **51**, **69**, **72**). In these cases, we reported the HRMS data of the $[M-HF]^+$ peaks as well as the low-res mass data of the $[M]^+$ peaks.

Mechanistically, the first step of the catalytic reaction is the electron-transfer from $[(Terpy)Cu(CF_2H)]$ to tetrahydropyran-4-carboxylic acid to generate Cu(II) species and a radical anion which undergoes decarboxylation to give alkyl radical. Is it possible that the electron-transfer step to $[(Terpy)Cu(CF_2H)]$ is more efficient than that to $[(bpy)Cu(CF_2H)_2]$?

Our response: This reviewer raises a very interesting point, which we have also been considering. Although we do not have strong evidence to rule out this possibility, the radical clock experiment suggests that this might not be the case. As shown in **Fig. 3C**, the direct decarboxylative difluoromethylation of compound **93** with either terpy or bpy as the ligand leads to the formation of the rearranged product **94** in high yields. This suggests that

$[(\text{terpy})\text{Cu}(\text{CF}_2\text{H})]$ and $[(\text{bpy})\text{Cu}(\text{CF}_2\text{H})_2]$ are both reactive enough to undergo single electron transfer with the redox-active esters.

C. Effect of ligands on the decarboxylative difluoromethylation of a radical clock

To better illustrate this point, the discussion has been modified in the revised manuscript as follows. “To investigate an alternative possibility that this ligand effect is due to the differences in reactivity of the $[\text{Cu}^{\text{I}}\text{-CF}_2\text{H}]$ species to undergo the single electron transfer event, we conducted the decarboxylative difluoromethylation of a radical clock substrate **93** (Fig. 3C). Direct decarboxylative difluoromethylation of **93** with either terpy or bpy as ligands under otherwise identical conditions both afforded the ring-opened product **94** in high yields along with trace amounts of the unrearranged product **95**. Although further experiments are necessary for a detailed comparison of the reactivity of different $[\text{Cu}^{\text{I}}\text{-CF}_2\text{H}]$ species, these results support that both $[(\text{terpy})\text{Cu}^{\text{I}}\text{-CF}_2\text{H}]$ and $[(\text{bpy})\text{Cu}^{\text{I}}\text{-CF}_2\text{H}]$ species were reactive enough to undergo the single electron transfer events to generate the corresponding benzylic radical, which rearranged to a primary alkyl radical, affording **94** as the major product.”

Considering the importance of the difluoromethyl group for the medicinal chemists and the field of pharmaceutical/argochemical industry, I suggest to accept it with minor revision.

Our response: Once again, we thank the referee for carefully evaluating our work and pointing out several issues that we should have clarified in the original manuscript.

Reviewer: 2

The authors describe a very interesting approach to a Cu-catalyzed carbo-difluoromethylation of various alkenes, which is in fact, a type of three-component reaction that involves two sequential Csp³-Csp³ bond formation initiated by a regioselective radical relay from secondary/tertiary alkyl radical to the alkene substrate. The driving force for the regioselectivity of this process is the formation of the most stable relayed radical, such as for instance the benzylic one, and the use of the appropriate ligands for the catalyst. The mechanistic conclusions are supported by both experimental and computational studies (DFT). Considering the complexity of the reaction (two C-C bond formations), its regioselectivity and the broad substrate scope and functional groups tolerance (including amine and hydroxyl groups), the yields of the large number of products synthesized are impressive. Therefore, this difluoromethylation may be useful for the synthesis of various carbo-CF₂H-containing derivatives, relevant for medicinal chemistry, in good yields. The manuscript is well organized and written. Based on the efficiency of this unique process and its high potential for the synthesis of new CF₂H-containing bioactive compounds, I suggest publication in Nature Communications subject to some minor revisions.

Our response: We thank the referee for valuable time spent evaluating this manuscript and the very supportive comments.

1. In the introduction (page 1 line 23) regarding the statement that CF₂H “act as a lipophilic hydrogen bond donor and thus a bioisostere for hydroxyl, amino, or thiol groups”, more accurately, it should be noted that for thiol, both groups, i.e. SH and CF₂H exhibit similar lipophilicities and hydrogen bond donating capabilities (J. Med Chem. 2019, 62, 5628-5637). There is a lack of appropriate references and examples for the increased implementation of this bioisosterism in medicinal chemistry. As in most cases, this is actually the rationale for the increased interest in the development of new synthetic approaches for difluoromethylation, and therefore, it is important to add a few of the recent reports on the utilization of the CF₂H group as a H-bond donor in drug discovery (see for instance ACS Med. Chem. Lett. 2018, 9(2), 143-8.; J. Med. Chem. 2018, 61, 10084-105; Eur. J. Med. Chem. 2018, 143, 473-90.; Steroids 2019, 151, 108469.

Our response: We thank the reviewer for pointing out these very relevant references on the importance of the difluoromethyl group in medicinal chemistry. All of these references have been properly cited in the revised manuscript (references 17-20).

In addition, we also thank the referee for noting that CF₂H groups and thiol groups exhibit similar lipophilicities and hydrogen bonding abilities. Although we tend not to discuss in very detail on the properties of CF₂H group in the introduction, we have revised the following sentence to avoid the misunderstanding that might be caused by our introduction:

“due to its unique ability to act as a **possible** lipophilic hydrogen bond donor and thus a **potential** bioisostere for hydroxyl, amino, or thiol groups”

2. Page 6 Table 2: a) the molecular structures as well as the font size of the atoms should be increased (the sizes in fig 4 are good enough).

Our response: We agree with the referee that the font size was a little bit small in the original manuscript. We have increased the font size of the atoms in the revised manuscript.

b) The diastereomeric ratio of compound 37 is missing. According to its spectrum in the SI it also seems to be of ca 1.2:1.

Our response: We thank the referee for pointing out this mistake. The d.r. ratio for compound 37 (1.2 : 1) has been added in Table 2.

c) More importantly, what is the rationale for the synthesis of compounds 54-57 in terms of drug design (or discovery)? These transformations, which describe a replacement of the carboxylic acid group in several drugs, by the aryl-difluoromethyl group, does not fall in any category of structure activity relationship studies nor in bioisosterism study. Thus the rationale for the synthesis of these specific modified drugs should be given.

Our response: For these studies, we would like to show that this carbo-difluoromethylation reaction can not only be applied to simple carboxylic acids, but also to the late-stage functionalization of complex molecules. This could also allow for the rapid construction of complex CF₂H molecules from readily available feedstocks. We chose oleanolic acid and glycyrrhetic acid since they are real drug molecules and contain sensitive functional groups, including unprotected alcohols and conjugated ketones, that were not evaluated in the functional group tolerance studies.

The following discussion has been revised in the manuscript to better present the rationale of this choice. “To further showcase the synthetic utility of this carbo-difluoromethylation protocol for the construction of complex alkyl-difluoromethanes, we sought to apply it to the late-stage modification of pharmaceuticals and natural products.”

3. Page 8 Table 3: the molecular structures as well as the font size of the atoms should be increased as mentioned above. The rationale for the replacement of the hydroxyl group in Combretastain by the CF₂H group is clear in terms of bioisosterism and is a very nice example for the potential implementation of the new reaction.

Our response: Once again, we agree that the original font size was small, and we have adjusted the size in the revised manuscript accordingly.

4. Page 12, Methods: Can the reaction be successfully performed only in a glovebox? If simple Schlenck cannot be used, this seems as a drawback of the reaction, as most organic chemists do not use glovebox for their syntheses.

Our response: The referee has raised a very interesting point. The main reason we set up this reaction in a glove box is due to the hygroscopicity of (DMPU)₂Zn(CF₂H)₂, which is kept in a freezer under an inert atmosphere for long-term storage. In addition, the copper(I) salts used in

this study are also stored in glovebox to prevent oxidation. To address the drawback pointed out by the reviewer, we have conducted the experiment without the glovebox, and a similar yield was achieved (vide infra).

This paragraph has been included in the revised supporting information: In open air, (DMPU)₂Zn(CF₂H)₂ (2.11 g, 5 mmol) was quickly dissolved in DMSO to make a stock solution of (DMPU)₂Zn(CF₂H)₂ (0.25 M). This solution was stored in a -20 °C freezer and no significant decomposition of the solution was observed by ¹⁹F NMR over the period of a month. In open air, a 4 mL oven-dried vial (A) equipped with a stir bar was charged with CuCl (30 mol %, 7.5 mg) and the RAE (11, 0.25 mmol, 1.0 equiv.). The vial was closed with a PTEE septum cap and wrapped with electrical tape. The vial was evacuated, backfilled with argon on a Schlenk line (three cycles), and was then added DMSO (200 μL). A different vial (B) charged with terpyridine (0.8 equiv., 47 mg) was evacuated and backfilled with argon on the Schlenk line (three cycles). The stock solution of (DMPU)₂Zn(CF₂H)₂ (800 μL) was then added to vial B using a syringe and mixed for 1 min to generate (terpy)Zn(CF₂H)₂ in-situ. The resulting solution was then added to vial A along with styrene (0.75 mmol, 3.0 equiv.) using a syringe. The resultant mixture was stirred at room temperature for 12 h. The yield was determined by ¹⁹F NMR using 1-fluoro-3-nitrobenzene as the internal standard (74%).

Once again, we thank the referee for the supportive comments and the great suggestions, providing us with the opportunity to further discuss many important points in our revised manuscript.

Reviewer: 3

Liu and coworkers present an interesting three component difluoromethylation of alkenes for the efficient synthesis difluoroalkylation products, which was achieved by copper catalyzed radical relay process. Compared to the extensively study on the difluoromethyl radical addition to alkenes, the current reaction proceeds a novel carbon radical coupled with Cu(II)-CHF₂ to generate sp³-sp³ C-C bond. As described in manuscript, broad substrate scope was observed to give products with moderate to good yields. More interesting, the ligand effect on the radical coupling was investigated from experimental and DFT calculation, and ligand terpy exhibited much better reactivity than bpy, where the reaction of cationic (terpy)Cu(II)CHF₂ with radical have lower energy barrier than neutral (bpy)Cu(II)CHF₂. Overall, this is a very nice study in the field, and the current manuscript is deserved to be publication in Nature Communication with minor revision.

Our response: We thank the referee for the valuable time spent reviewing our manuscript and the great support of this work.

(1) For the optimizing reaction condition, the full picture of the reaction should be provided. For instance, based on the reaction in Fig. 3B, the alkyl radical generated from decarboxylation is possibly trapped by (terpy)Cu(II)CHF₂. Thus, the side product should be addressed in Table 1, which is helpful to readers.

Our response: We thank the referee for this great suggestion. The formation of the side product, **12a**, formed under the optimized conditions has been included in the revised Table 1 (also shown below). The following sentence has also been included in the main text: In addition, **12a**, derived from the direct capturing of the alkyl radical by the [Cu^{II}-CF₂H] species, was formed in a 11% yield under the optimized conditions.

(2) For the ligand effect, another possibility should be discussed, where the transmetalation step of Cu(I) with LZn(CHF₂)₂ might also affect by ligand.

Our response: The referee makes a very interesting point, which we did not consider in the previous submission. We agree with the reviewer that Cu(I) salts could have different transmetalation rates with LZn(CF₂H)₂ bound with either bpy or terpy. Yet, the data in Fig. 3C suggests that with either bpy or terpy as the ligands, the redox active ester **93** could undergo an efficient decarboxylation reaction to generate the benzylic radical. Therefore, it is reasonable to postulate that the decarboxylative difluoromethylation of **89** (Fig. 3B) also leads to the formation of a benzylic radical with either terpy or bpy as the ligand. The formation of radical dimerized product when bpy was used as the ligand also supports this postulation (Fig. S3). Yet, the

dramatic differences in the yields of the difluoromethylated product **90** with these two different ligands suggest the different outcome of the benzylic radicals in these two systems. Although we cannot rule out the possibility that this is due to the different transmetalation rates, the DFT calculation results are more consistent with the effect of ligands on the reactivity of Cu^{II} species.

B. Effect of ligands on the decarboxylative difluoromethylation reactions

C. Effect of ligands on the decarboxylative difluoromethylation of a radical clock

That being said, in order to take into account the possibility of the transmetalation effect on the reactivity raised by the reviewer, the discussion has been revised as follows in the revised manuscript: “Overall, these DFT calculations suggest that the success of the tridentate ligand, terpy, in the carbo-difluoromethylation reactions was largely due to a rapid combination of the secondary benzylic radicals with the Cu^{II} intermediates as well as the facile reductive elimination of the Cu^{III} species, although other factors including the differences of the transmetalation rates could not be ruled out.”

(3) For the reaction in Fig. 3C, the reaction gave the similar yields with terpy and bpy, wherein the primary carbon radical was involved. The possible reason should be given in the main text.

Our response: Our current hypothesis for this result is due to the small steric hindrance when a primary radical reacts with a [Cu^{II}-CF₂H] species. The following discussion has been included in the revised main text: “The high yields for the formation of **94** with either ligand could be explained by the low steric interaction when a primary radical reacts with the [Cu^{II}-CF₂H] species.”

Once again, we thank the referee for carefully evaluating our work and providing us with the opportunity to further revise our work.

Reviewers' Comments:

Reviewer #1:

Remarks to the Author:

The authors addressed my questions and I suggest to accept it.

Reviewer #2:

Remarks to the Author:

My comments were all taken into account. so, I recommend acceptance as is.

Reviewer #3:

Remarks to the Author:

Author have well answered the questions raised by this referee. Thus, the current manuscript is strongly recommended to publication in Nature Communication as it is.